# Molecular Basis of the Pathogenic Mechanism Induced by the m.9191T>C Mutation in Mitochondrial *ATP6* Gene

**DOI:** 10.3390/ijms21145083

**Published:** 2020-07-18

**Authors:** Xin Su, Alain Dautant, François Godard, Marine Bouhier, Teresa Zoladek, Roza Kucharczyk, Jean-Paul di Rago, Déborah Tribouillard-Tanvier

**Affiliations:** 1Institut de Biochimie et Génétique Cellulaires, Université de Bordeaux, 1 Rue Camille Saint-Saëns, 33077 Bordeaux, France; xin.su@ibgc.cnrs.fr (X.S.); a.dautant@ibgc.cnrs.fr (A.D.); francois.godard@ibgc.cnrs.fr (F.G.); marine.bouhier@ibgc.cnrs.fr (M.B.); jp.dirago@ibgc.cnrs.fr (J.-P.d.R.); 2Institute of Biochemistry and Biophysics, Polish Academy of Sciences, 02-106 Warsaw, Poland; teresa@ibb.waw.pl (T.Z.); roza@ibb.waw.pl (R.K.)

**Keywords:** mitochondrial disease, *ATP6* gene, ATP synthase, genetic suppressors, yeast, mitochondrial DNA mutation, proton conduction, m.9191T>C

## Abstract

Probing the pathogenicity and functional consequences of mitochondrial DNA (mtDNA) mutations from patient’s cells and tissues is difficult due to genetic heteroplasmy (co-existence of wild type and mutated mtDNA in cells), occurrence of numerous mtDNA polymorphisms, and absence of methods for genetically transforming human mitochondria. Owing to its good fermenting capacity that enables survival to loss-of-function mtDNA mutations, its amenability to mitochondrial genome manipulation, and lack of heteroplasmy, *Saccharomyces cerevisiae* is an excellent model for studying and resolving the molecular bases of human diseases linked to mtDNA in a controlled genetic background. Using this model, we previously showed that a pathogenic mutation in mitochondrial *ATP6* gene (m.9191T>C), that converts a highly conserved leucine residue into proline in human ATP synthase subunit *a* (*a*L222P), severely compromises the assembly of yeast ATP synthase and reduces by 90% the rate of mitochondrial ATP synthesis. Herein, we report the isolation of intragenic suppressors of this mutation. In light of recently described high resolution structures of ATP synthase, the results indicate that the m.9191T>C mutation disrupts a four α-helix bundle in subunit *a* and that the leucine residue it targets indirectly optimizes proton conduction through the membrane domain of ATP synthase.

## 1. Introduction

Mitochondria provide aerobic eukaryotes with ATP though the process of oxidative phosphorylation (OXPHOS) [1]. In this process, reducing equivalents released by fatty acid and carbohydrate oxidation are transferred to oxygen typically by four complexes (I–IV) embedded in the mitochondrial inner membrane, and this is coupled to a pumping of protons from the mitochondrial matrix to the intermembrane space (IMS). The resulting transmembrane electrochemical ion gradient, also called proton-motive force (pmf), is used by the ATP synthase (or complex V) to produce ATP from ADP and inorganic phosphate (Pi) [2]. 

The ATP synthase organizes into a membrane-extrinsic domain (F_1_) and a domain (F_O_) largely anchored in the inner membrane [3,4,5]. The subunit *a* and a ring of identical subunits *c* (*c*-ring) move protons through the F_O_, which is coupled to rotation of the *c*-ring and ATP synthesis in the F_1_ [4,6,7,8,9]. Devastating human neuromuscular disorders like neuropathy ataxia retinitis pigmentosa (NARP) and maternally inherited Leigh syndrome (MILS) have been associated to specific mutations in the mitochondrial *ATP6* gene, which encodes the subunit *a* [10,11,12]. In most cases these mutations co-exist with wild type mitochondrial DNA molecules (heteroplasmy), which makes it difficult to know precisely how they impact ATP synthase from patient’s cells and tissues. As we have shown, the yeast *Saccharomyces cerevisiae* is a convenient model for the study of these mutations [13,14,15,16,17,18,19,20]. Its mitochondrial genome can be modified [21], and owing to the instability of heteroplasmy in this organism [22], it is possible to obtain homoplasmic clones where all the mtDNA molecules carry the same mutation. We found in this way that an equivalent of the m.9191T>C mutation identified in patients presenting with MILS [23] severely compromises the assembly/stability of yeast ATP synthase and reduces by 90% the rate of mitochondrial ATP synthesis [18]. This mutation converts a highly conserved leucine residue into proline at amino acid position 242 of yeast subunit *a* (*a*L242, which corresponds to *a*L222 in the human protein) [18]. Subsequent to this work, high-resolution structures of complete yeast ATP synthase were reported [6,10], which reveals that *a*L242 belongs to an α-helix at the proximity of the C-terminus of subunit *a* (see below). Due to the high propensity of proline residues to break α-helices, this can explain the detrimental consequences of the leucine-to-proline change on the stability/assembly of subunit *a* (see below).

As many amino acid residues can be accommodated well in α-helices, we hypothesized from its high evolutionary conservation that *a*L242 may have another role than simply allowing the C-terminal domain of subunit *a* to fold as an α-helix. This hypothesis was investigated herein through the isolation of intragenic suppressors from the *a*L242P mutant. In this way, we found that ATP synthase assembly and stability were fully restored when *a*P242 was converted into serine or threonine. However, interestingly, the activity of ATP synthase was far from being recovered completely (50–60% vs wild type yeast). Based on these findings, and in light of the reported atomic structures of yeast ATP synthase [6,10], we propose that *a*L242 optimizes the interaction of protons released from the *c*-ring with a nearby aspartic acid residue in subunit *a* (*a*D244) that has been proposed to be important for the activity of F_O_.

## 2. Results

### 2.1. Isolation of Revertants from the Mutant aL242P 

Yeast subunit *a* (also referred to as subunit *6* or Atp6) is synthesized as a pre-protein, the first ten residues of which are removed during ATP synthase assembly [24]. The leucine residue at position 222 of human subunit *a* that is changed into proline by the m.9191T>C mutation corresponds to *a*L242 in the mature yeast protein (252 in unprocessed yeast subunit *a*) (Table 1). As reported [18], a yeast model of this mutation was constructed by replacing the TTA leucine codon 252 with the CCA proline codon. This mutation considerably compromises the ability of yeast to grow on non-fermentable carbon sources (like glycerol) that require a functional ATP synthase to be metabolized (Figure 1A,B). Revertants from the *a*L242P mutant showing a much better respiratory growth were isolated using a previously described procedure ([25], see also Materials and Methods). To this end, 40 subclones of the *a*L242P mutant were isolated on rich glucose (YPGA) plates, individually cultured in liquid 10% glucose, and spread on rich glycerol (YPGlyA) plates (10^8^ cells/plate from each *a*L242P subclone). Respiratory competent clones emerged from the glycerol plates at a 10^-7^ frequency (i.e., 10 clones per plate on average). Forty-five revertant clones (maximum two per selection plate) were retained and genetically purified by subcloning on glucose plates. This procedure ensures that most of the isolates are genetically independent (i.e., not brothers or sisters produced by mitotic divisions). The *ATP6* genes of the retained clones were amplified and entirely sequenced. In 31 revertants, the CCA proline codon 252 was replaced with TCA serine codon. In 14 others, the proline codon was replaced with a threonine codon (ACA or CTA, see Table 1). The absence of true back mutation restoring the original leucine TTA codon 252 is because two nucleotide changes were introduced to convert it into a proline codon (CCA). Spontaneous reversion of CCA to TTA would therefore be an extremely rare event with a frequency <10^−12^ (thousands of selection plates would be needed to find it). The revertants with a serine or threonine codon at position 252 will be designated below as *a*L242S and *a*L242T respectively, to indicate the amino-acid changes relative to the wild type sequence of mature yeast subunit *a*. Theoretically, three other codons specifying arginine (CGA), alanine (GCA), or glutamine (CAA) could have been derived by a single nucleotide substitution from the mutant proline codon CCA (Table 1). Possible explanations for the absence of these reversions in the analyzed clones are provided below.

The *a*L242S and *a*L242T strains grew well on solid (Figure 1A) and in liquid (Figure 1B) glycerol media, almost as wild type (WT) yeast. This does not imply that ATP synthase assembles and functions normally in these strains, because the rate of mitochondrial ATP synthesis needs to be decreased by at least 80% to see obvious defects in the respiratory growth of yeast [16,18,26]. As reported [18], the *a*L242P strain has a somewhat higher tendency to produce cells with large deletions in mitochondrial DNA (ρ^−^/ρ^o^) compared to the WT (15% vs. <1%), which is a phenotype frequently observed in strains with very severe defects in ATP synthase [27,28]. Consistent with their good growth on glycerol and thus certainly with a significant recovery of ATP synthase function (beyond the 20% threshold), the *a*L242S and *a*L242T strains had a good genetic stability (<1%) (Table 2).

### 2.2. Assembly of OXPHOS Complexes

To evaluate directly the influence on ATP synthase of the suppressor amino acid changes *a*L242S and *a*L242T, mitochondria were isolated from cells grown in a medium (YPGalA) containing galactose, which is a fermentable substrate that does not elicit repression of mitochondrial proliferation as glucose does. For comparison, mitochondria were prepared also from WT yeast and the *a*L242P mutant. Mitochondrial samples were solubilized with digitonin and run in a non-denaturing polyacrylamide gel (BN-PAGE). The ATP synthase was in-gel visualized by its ATPase activity and Coomassie blue staining. In the conditions used, ATP synthase monomers were much more abundant than dimers of this enzyme (Figure 2A). As we already reported [18], fully assembled ATP synthase was in very reduced amounts in *a*L242P mitochondria. Samples from the strains *a*L242S and *a*L242T (with either a CTA or ACA threonine codon) had a good content in ATP synthase. To better quantify the amounts of ATP synthase, we evaluated by SDS-PAGE the steady state levels of two of its subunits (the α-F_1_ subunit (Atp1) and subunit *a* (Atp6)). After their electrophoretic separation, the proteins were transferred to a nitrocellulose membrane and probed with antibodies against Atp1, Atp6, and porin (a protein of the outer mitochondrial membrane used for normalization of Atp1 and Atp6). The Atp6 and Atp1 proteins showed an almost normal accumulation in the samples prepared from the *a*L242S and *a*L242T strains (Figure 2B and Figure 3A). It can then be inferred, also considering the BN-PAGE data, that the presence of a threonine or serine residue at position 242 of subunit *a* has a minimal, if any, impact on the assembly and stability of ATP synthase.

As was observed in many yeast ATP synthase defective mutants [13,14,15,16,29], the severe consequences of the *a*L242P mutation result in a diminished content in Complex IV (20% vs WT) [18], which is the last complex of the respiratory chain that transfers electrons from cytochrome *c* to oxygen. This effect was not observed in mutants with proton leaks through the F_O_ indicating that it is the activity of F_O_ that modulates Complex IV biogenesis, presumably as a means to co-regulate mitochondrial electron transfer and ATP synthesis activities [30]. It was argued that the increase in the mitochondrial electrical potential (ΔΨ) consecutive to a diminished F_O_ activity impairs some ΔΨ-dependent step in the assembly of Complex IV. In addition to the restoration of a good content in ATP synthase, the severe drop in Complex IV induced by the *a*L242P mutation was also largely suppressed in the *a*L242S and *a*L242T strains. In the BN-gels shown, the Complex IV was mostly associated with dimers of Complex III (III_2_-IV_2_ and III_2_-IV_1_), as revealed by WB with antibodies against the Cox2 subunit of complex IV, and by the in-gel Complex IV activity (Figure 2C). 

### 2.3. Mitochondrial Respiration and ATP Synthesis

Oxygen consumption was measured in mitochondria with NADH as an electron donor, alone (basal or state 4 respiration), and after successive additions of ADP (state 3, phosphorylating conditions) and CCCP (uncoupled, maximal respiration). Consistent with our previous study [18], these activities were strongly reduced in mitochondria from the *a*L242P mutant, by 85% both at state 3 and in the presence of CCCP compared to WT (Table 2). Consistent with the large drop in Complex IV in mutant *a*L242P [18], the activity of this complex measured with ascorbate/TMPD as an electron source was reduced in a similar proportion (15% of WT) (Table 2). The mitochondria from strains *a*L242S and *a*L242T respired much better, albeit not as well as in the WT mitochondria (55–65% with NADH at state 3 and in the presence of CCCP, and 50–75% with ascorbate/TMPD). 

We next measured the rate of ATP synthesis, using NADH as an electron donor and in the presence of a large excess of external ADP so as to maximize this activity. These measurements were made in the presence and absence of oligomycin (a specific inhibitor of ATP synthase), to determine the part of this activity specifically due to the ATP synthase. As previously reported [18], the rate of F_1_F_O_-mediated ATP synthesis was very weak in *a*L242P mitochondria (10% of the WT) due to their poor content in ATP synthase and Complex IV (Table 2; Figure 2B,C). ATP was produced much more rapidly (50–60%) in the mitochondria from the *a*L242S and *a*L242T strains (Table 2; Figure 3B). Importantly, the yield in ATP per electron transferred to oxygen was almost unaffected in the revertants (Table 2; Figure 3C). Thus, while the presence of a serine or threonine residue at position 242 has almost no, if any, effect on the abundance of fully assembled ATP synthase, these residues significantly slow down the rate at which ATP synthase produces ATP (see below for a possible explanation). 

### 2.4. Mitochondrial ATP Hydrolysis

We assessed the functioning of ATP synthase in the reverse mode by measuring the rate of ATP hydrolysis in mitochondria not osmotically protected so as to relax ATP synthase from any membrane potential that would limit its ATP hydrolytic activity, and this was done at pH 8.4 to avoid any binding to F_1_ of its natural inhibitor IF1 [17]. These measurements were made in the presence and absence of oligomycin. In the presence of oligomycin, this activity is normally inhibited by 80–90% which is because the *c*-ring cannot rotate and this blocks F_1_-mediated ATP hydrolysis (Table 2). The remaining oligomycin-insensitive activity is due to other ATPases present in the organelle. The mitochondrial ATPase activity in *a*L242P mitochondria was only modestly diminished, which is not surprising owing to the capacity of F_1_ to assemble while the F_O_ cannot. However, being relaxed from the F_O_ due to the lack in subunit *a* in the *a*L242P mutant, F_1_ can hydrolyze ATP in the presence of oligomycin. Mitochondria from the *a*L242S and *a*L242T strains had a good ATPase activity and this activity was normally inhibited by oligomycin, from which it can be deduced that the F_1_ and F_O_ are functionally coupled in these strains.

### 2.5. Mitochondrial Membrane Potential

We further investigated ATP synthase functionality in the *a*L242S and *a*L242T strains by membrane potential (ΔΨ) measurements, using Rhodamine 123, a cationic dye whose fluorescence is quenched when ΔΨ increases. For evaluating ΔΨ-consumption by the ATP synthase, the mitochondrial membrane was first energized by feeding the respiratory chain with electrons from ethanol, and a small amount of ADP was then added. The addition of ADP results in a transient fluorescence increase due to proton reentry through the F_O_F_1_-ATP synthase until complete phosphorylation of the added ADP (Figure 4A). Due to their very weak capacity to produce ATP, the mitochondria from the *a*L242P mutant responded extremely poorly to the addition of ADP. Those from the *a*L242S and *a*L242T strains responded much better but the return to the ethanol-induced ΔΨ value (after complete phosphorylation of the added ADP) took a longer time than in WT mitochondria (Figure 4A). This is consistent with the slowing down of ATP synthase when *a*L242 is replaced with either serine or threonine (see above). The experiment was pursued by adding KCN, which results in ΔΨ collapse due to respiratory chain inhibition. The loss in ΔΨ was partial in WT, *a*L242S and *a*L242T mitochondria, which is because the ATP produced during phosphorylation of the externally added ADP is now hydrolyzed by the ATP synthase coupled to F_O_-mediated proton transport outside the organelle, as evidenced by the loss of the remaining ΔΨ after a further addition of oligomycin (Figure 4A). In *a*L242P mitochondria, the ethanol-induced ΔΨ was totally lost in one single rapid phase after the addition of KCN, which further reflects their extremely poor capacity to produce ATP (Figure 4A). 

We further tested ATP synthase reverse functioning by providing the mitochondria with externally added ATP. The mitochondria were first energized with ethanol and then treated with KCN, which results is a ΔΨ collapse that removes IF1 from F_1_. Less than one minute later, well before IF1 rebinding [31], the ATP was added. A large oligomycin-sensitive ΔΨ-variation was, as expected, observed in the WT mitochondria (Figure 4B). Although the *a*L242P mitochondria still had a good F_1_-mediated capacity to hydrolyze ATP (see above), almost no ΔΨ variation was observed due to the absence in these mitochondria of a functional F_O_. On the contrary, mitochondria from the *a*L242T and *a*L242S mutants showed a good oligomycin-sensitive proton pumping activity, which further illustrated their capacity to assemble an active F_O_ (Figure 4B).

### 2.6. Structural Modeling

Mitochondrial ATP synthase subunit *a* is an intrinsic membrane protein made of two domains, a N-term transmembrane α helix (*a*H1) a C-term 4-helix bundle (*a*H3-*a*H6) horizontally wrapped around the *c*-ring rotor. These two domains are linked by an amphipathic α-helix (*a*H2) lying along the matrix surface of the mitochondrial inner membrane. Two charged residues (*a*R176 and *c*E59 in yeast ATP synthase) near the middle of the membrane, at the interface between the *c*-ring and subunit *a*, are known for a long time to be essential for moving protons through the F_O_ coupled to rotation of the *c*-ring [6,9]. Three other charged residues (*a*E162, *a*D244 and *a*E223) would serve as intermediate proton binding sites [32] (Figure 5B,C).

The leucine residue at position 242 (222 in humans) that is targeted by the m.9191T>C pathogenic mutation is highly conserved in a large panel of evolutionary distant species or replaced by other aliphatic side chain residues (I, V) (Figure 5A). In *S. cerevisiae* cryo-EM structure (Pdb_id:6b8h, 3.4 Å resolution [6), *a*L242 is located near the C-terminus of *a*H6, at the heart of the 4-helix bundle, quite far away from the *a*/*c* interface (Pdb_id:6b8h, 3.4 Å resolution) (Figure 5D). With a less bulkier side chain than leucine, a proline residue at position 242 possibly reduces the stability of this helix bundle and as a helix breaker it may compromise the winding of the last turn of the C-terminal α-helix. The C-terminal domain of subunit *a* was shown to interact with a protein called Atp10 that chaperones subunit *a* or protects it from degradation until incorporation into ATP synthase [33,34]. Interestingly, very close to position 242, an alanine-to-valine change (*a*A239V) was shown to partially suppress a lack in Atp10 [35]. The *a*L242P mutation possibly compromises the interaction between subunit *a* and Atp10. These defective interactions are likely responsible for the degradation of the mutated subunit *a*, as usually observed when subunit *a* cannot assemble properly [15,36]. 

As shown above, the folding and/or stability of subunit *a* within ATP synthase was fully restored upon reversion of *a*P242 into serine or threonine. These residues are known to be accommodated well in α-helical secondary structures especially when located close to their C-terminus [37]. Possibly the hydroxyl group of serine and threonine establishes a hydrogen bond with the amide group of *a*N129 and this restores the stability of the subunit *a* 4-helix bundle (Figure 5E). The slow-down of ATP synthase in the *a*L242S and *a*L242T mutants (50–60% vs WT, see above) is possibly caused by a local structural change that impairs the proton conduction activity of *a*D244 (see below). 

## 3. Discussion

At the time we found that an equivalent of the m.9191T>C mutation of the mitochondrial *ATP6* gene strongly compromises (by 90%) yeast ATP synthase assembly [18], complete structures of this enzyme were not yet available, and it was therefore difficult to understand the molecular basis of this effect. In the recently described high-resolution structures of yeast ATP synthase [9], the leucine-to-proline change induced by this mutation at position 242 of subunit *a* (222 in the human protein) is within the last helical domain of subunit *a* (*a*H6) (Figure 5). This domain is at the interface between subunit *a* and the *c*-ring and harbors residues (*a*E223 and *a*D244) that presumably act as proton binding sites for moving protons from the intermembrane space to the mitochondrial matrix (Figure 5B,C) [6]. Due to the well-known difficulty of proline residues to be accommodated within alpha helical structures, the *a*L242P change possibly prevents the subunit *a* to adopt a stable conformation and makes it prone to proteolytic degradation, as was observed in many yeast mutants where the subunit *a* cannot be stably incorporated into ATP synthase [13,14,15,17,18,20]. The *a*L242 residue does not face the *c*-ring. Rather, it is within a bundle of four alpha helices (*a*H2, *a*H3, *a*H4, and *a*H6) backwards the *a*/*c* interface in proximity to other hydrophobic residues, that are presumably important for the packing and stability of these four helices. While hydrophobic aliphatic side chain residues (L, I, V) predominantly occupy this position in a vast panel of evolutionary distant organisms (Figure 5A), the present study reveals that polar uncharged serine and threonine residues at position 242 enable an effective assembly of ATP synthase as in wild type yeast. However, the rate of mitochondrial ATP synthesis, with these residues, is slowed down (by 50%) while the yield in ATP per electron transferred to oxygen is mostly unaffected. It can be deduced that F_O_-mediated proton conduction is less rapid with *a*S242 and *a*T242 than in the wild type enzyme. With a side-chain unable to exchange protons and located outside the *n*-side hydrophilic pocket that connects the *c*-ring and the mitochondrial matrix, the wild type *a*L242 residue is certainly not directly involved in moving protons. It can however favorably influence this pocket by enabling the nearby *a*D244 residue to adopt a position that optimizes its proton conduction activity. With *a*S242 and *a*T242, this aspartate residue may interact less efficiently with protons released by the *c*-ring.

While it may seem obvious that breaking *a*H6 with a proline residue is responsible for the pathogenicity of m.9191T>C, we did not observe defects in yeast ATP synthase assembly with equivalents of three other pathogenic subunit *a* mutations that lead also to the introduction of a proline residue in an alpha helix (aS240P (m.9185T>C) in *a*H6 [18]; *a*L173P (m.8993T>C) in *a*H5 [14]; and *a*L237P (m.9176T>C) in *a*H6 [16]) (Figure 5G). We reported, before the advent of complete high-resolution structures of ATP synthase, that these three mutations partially compromise the functioning of ATP synthase, with 30–50% drops in the rate of mitochondrial ATP synthesis. State 3 respiration was found unaffected in the *a*S240P mutant, indicating a less efficient coupling of electron transfer to ATP synthesis [18]. The *a*S240P change possibly results in a slight distortion/displacement of the C-terminal domain of *a*H6 that partially compromises the sealing at the center of the membrane of the two hydrophilic pockets between the *c*-ring and subunit *a* (Figure 5G). As a result of this, a proton shortage between the two pockets may be responsible for the reduced yield in ATP synthesis in this mutant. As in the *a*L242S and *a*L242P mutants, proton conduction within the F_O_ was found to be slower in the *a*L173P [14] and *a*L237P [16] strains vs WT yeast without any defect in the incorporation/stability of subunit *a* within ATP synthase and yield in ATP per electron transferred to oxygen. Another mutation found in patients (m.9176T>G) that converts *a*L237 into an arginine residue (*a*L237R) was found to dramatically compromise the assembly of yeast ATP synthase due to electrostatic and steric hindrance with the catalytic *a*R176 residue [15]. The isolation of revertant strains from the *a*L237R mutant led us to conclude that *a*L237, in addition to a role in properly shaping subunit *a,* is functionally important by optimizing the interaction of *a*R176 with the catalytic residue of subunit *c* (*c*E59) [14], similarly to the conclusions drawn here about the importance of *a*L242 to optimize proton conduction within the *n*-side pocket.

Theoretically, three other codons specifying arginine (CGA), alanine (GCA), or glutamine (CAA), could have been derived by a single nucleotide substitution from the mutant proline codon CCA at position 252 of the *ATP6* codon sequencing (corresponding to amino acid position 242). Since a quite large number of genetically independent revertant strains from the *a*L242P mutant were characterized, the absence of these codons is intriguing. The absence of the arginine CGA codon is likely due to the fact that it is never used in yeast mitochondria, whereas the two others (CAA and GCA) are used quite frequently [38,39]. If transversions, which are known to be less frequent than transitions, are required to change CCA into GCA or CAA, this is likely not the reason why they were not selected. Indeed, two transitions and one transversion are responsible for those suppressors that convert *a*L242 into serine or threonine. It is thus a reasonable possibility that *a*Q242 and *a*A242 are not compatible with ATP synthase function despite their strong propensity to induce alpha helical structures. Consistently, these residues are never found at the corresponding position of subunits *a* from evolutionary distant species (Figure 5A). Contrary to *a*S242 and *a*T242, the small aliphatic side chain of alanine (a single methyl group) cannot form a hydrogen bond with *a*N129, and such an interaction with *a*Q242 seems unlikely because its side chain is too long (Figure 5F). However, this might not be the sole reason for the putative incompatibility of *a*A242 and *a*Q242 with ATP synthase assembly and function. Because of its small side chain size, *a*A242 will create a cavity between *a*H2-4 and *a*H6 that may prevent a correct folding or stability of the subunit *a*, and *a*Q242 (which has one more aliphatic carbon compared to the wild type *a*L242 residue) would likely be responsible for a steric hindrance within the 4-helix bundle of subunit *a* (Figure 5F).

In previous studies of yeast models of other pathogenic subunit *a* mutations (m.G8969G>A [20], m.8851T>C [17]), second-site reversions were found [40,41], meaning that ATP synthase function can be recovered while the original mutation is still present, which is an interesting observation that holds promise for the development of therapeutic molecules that can induce the beneficial structural changes that the genetic suppressors provoke. The present study reveals that an ATP synthase pharmacological targeting approach is a difficult perspective with the m.9191T>C mutation because its elimination seems to be the only way to recover a substantial bioenergetic capacity.

## 4. Materials and Methods 

### 4.1. Growth Media and Genotypes

The media used for growing yeast were: YPGA (1% Bacto yeast extract, 2% Bacto Peptone, 2% or 10% glucose, 60 mg/L adenine), YPGalA (1% Bacto yeast extract, 2% Bacto Peptone, 2% galactose, 60 mg/L adenine), and YPGlyA (1% Bacto yeast extract, 2% Bacto Peptone, 2% glycerol, 60 mg/L adenine). Solid media were obtained by adding 2% Bacto Agar (Difco, Becton Dickinson, Grenoble, France). The genotypes of the strains used in this study are listed in Table 3. Growth curves were established with the Bioscreen C^TM^ system.

### 4.2. Selection of Revertants from the Yeast Strain aL242P Mutant

The *a*L242P mutant (yeast strain RKY66) that carries an equivalent of the m.9191T>C mutation was subcloned on rich 2% glucose (YPGA) plates. Forty subclones were picked up and individually grown for two days in 10% glucose (YPGA). Glucose was removed from the cultures by two washings with water and 10^8^ cells from each culture were spread on rich glycerol (YPGlyA) plates. The plates were incubated at 28 °C for at least fifteen days. A maximum of two revertants per plate were retained for further analysis to ensure genetic independence of the isolates. The revertants were purified by subcloning on rich glucose (YPGA) plates and their *ATP6* gene was PCR-amplified with two primer pairs ATP6.1 (Forward, from nucleotide position –100 upstream of the *ATP6* start codon) 5′TAATATACGGGGGTGGGTCCCTCAC, ATP6.8 (Reverse, from nucleotide position +743 downstream of the *ATP6* start codon) 5′CTGCATCTTTTAAATATGATGCTG and ATP6.2 (Forward, from nucleotide position +337 downstream of the *ATP6* start codon) 5′GTATGATTCCATACTCATTTG, ATP6.10 (Reverse, from nucleotide position +218 downstream of the *ATP6* stop codon) 5′GGGCCGAACTCCGAAGGAGTAAG and then entirely sequenced.

### 4.3. Miscellaneous Procedures

Mitochondria enriched fractions were prepared by an enzymatic method and differential centrifugation as in [42] from yeast cells that have been grown until middle exponential phase (2–3 × 10^7^ cells/mL) in rich galactose (YPGalA) media at 28 °C. Protein concentration was determined by the Lowry method [43] in the presence of 5% SDS. Oxygen consumption rates were measured with a Clark electrode using 150 µg/mL of mitochondrial proteins in a respiration buffer (0.65 M mannitol, 0.36 mM ethylene glycol tetra-acetic acid, 5 mM Tris/phosphate, 10 mM Tris/maleate pH 6.8) as in [44]. The additions were 4 mM NADH, 12.5 mM ascorbate (Asc), 1.4 mM *N*,*N*,*N*′,*N*′-tetramethyl-p-phenylenediamine (TMPD), 150 µM ADP, and 4 µM carbonyl cyanide m-chlorophenylhydrazone (CCCP). Variations in transmembrane potential (ΔΨ) were evaluated with 150 µg/mL of mitochondrial protein in the respiration buffer by measurement of Rhodamine 123 (25 μg/mL) fluorescence quenching as described in [45]. The additions were 10 μL ethanol (EtOH), 75 μM ADP, 2 mM potassium cyanide (KCN), 0.2 mM ATP, 4 μg/mL oligomycin (oligo), and 4 μM CCCP. Variation of mitochondrial membrane potentials are shown as traces of fluorescence recorded by a SAFAS Monaco fluorescence spectrophotometer. ATP synthesis rate measurements were performed with 150 µg/mL of mitochondrial proteins in respiration buffer supplemented with 4 mM NADH and 1 mM ADP in a thermostatically controlled chamber at 28 °C, as described in [29]. Aliquots of 50 µL were withdrawn from the oxygraphy cuvette every 15 s and the reaction was stopped with 7% (*w*/*v*) perchloric acid, 25 mM EDTA. The samples were then neutralized to pH 6.8 by adding 2 M KOH/0.3 M MOPS and ATP was quantified using a luciferin/luciferase assay (ATPlite kit from Perkin Elmer) and the CLARIOstar microplate reader (from BMG LABTECH, Ortenberg, Germany). The participation of the F_1_F_O_-ATP synthase in ATP production was assessed by adding oligomycin (20 µg/mg of protein). ATPase activity measurements were performed with 150 µg/mL of mitochondrial proteins in an ATPase buffer (0.2 M KCl, 3 mM MgCl_2_, 10 mM Tris/HCl, adjusted to pH 8.4) using a described procedure [46]. The reaction was started by the addition of 50 µL 0.1 M ATP. After 2 min incubation, the reaction was stopped, and proteins were precipitated by adding TCA 100%. ATP hydrolysis was measured by following Pi release using ammonium molybdate solution spectrophotometric method at 610_nm_ [47]. The participation of the F_1_F_O_-ATPase in ATP hydrolysis was assessed by adding oligomycin (2 mg/mL). Blue native polyacrylamide gel electrophoresis (BN-PAGE) analyses were carried out as described previously [48]. Briefly, mitochondrial extracts solubilized with digitonin (the concentrations used are indicated in the legend of figures) were separated in a 3–12% acrylamide continuous gradient gel and complex IV and V complexes were revealed in gel by their activities as described [49], or by Western blot after transferred to polyvinylidene difluoride (PVDF) [50]. The in-gel complex IV and V activities were revealed using a solution of 5 mM Tris, 0.5 mg/mL diaminobenzidine, 0.05 mM cytochrome *c,* and a solution of 270 mM glycine, 35 mM Tris adjusted to pH 8.4, 14 mM MgSO_4_, 0.2% lead nitrate, and 8 mM ATP, respectively. SDS-PAGE analyses were performed according to [51]. Polyclonal antibodies against Atp6 (subunit *a*, a gift from J. Velours), and Atp1 (subunit α-F_1_, a gift from J. Velours) were used after 1:5000 dilution. Monoclonal antibodies against Cox2 (Molecular Probes, Eugene, OR, USA) subunit of complex IV was used after dilution 1:500. Membranes were incubated with peroxidase-labeled antibodies (Promega, Southampton, UK) at a 1:5,000 dilution and revealed with the ECL reagent of Pierce^TM^ ECL Western Blotting Substrate (ThermoScientific, Waltham, MA, USA). Immunological signal quantification was performed with ImageJ software [52]. 

### 4.4. Amino-Acid Alignments and Structural Modeling

Multiple sequence alignment of ATP synthase *a*-subunits of various origins was performed using Clustal Omega [53]. Molecular views of subunit *a* and *c*_10_-ring are drawn from the dimeric F_O_ domain of *S. cerevisiae* ATP synthase (pdb_id: 6b2z; 3.6 Å resolution [6]). All structure figures were drawn using the PyMOL molecular graphic system [54]. The *a*L242S/T point mutations were introduced using the mutagenesis feature in PyMOL selecting the side-chain rotamer not sterically colliding with neighboring residues. After a 180° rotation of the *a*Asn129 side chain, the amide nitrogen is in position to hydrogen bond with the hydroxyl groups of *a*Ser242 and *a*Thr242. 

### 4.5. Statistical Analysis

At least three biological and technical replicates were performed for all experiments. Statistical significance of the data was tested using unpaired *t*-test.

## Figures and Tables

**Figure 1 ijms-21-05083-f001:**
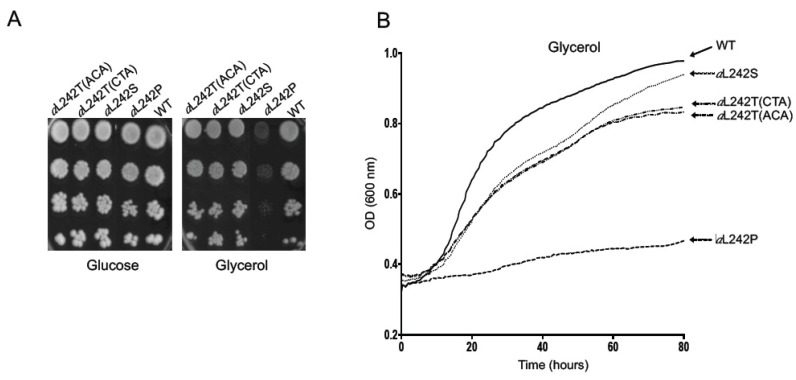
Respiratory growth of yeast strains. (**A**) Fresh glucose cultures of wild type yeast (*WT*), and the mutant strains *a*L242P, *a*L242T and *a*L242S were serially diluted and spotted onto rich glucose and glycerol plates. The shown plates were scanned after 4 days incubation at 28 °C. (**B**) Growth of the same strains in liquid glycerol medium monitored with the Bioscreen system over a period of 80 h during which cell densities (OD_600nm_) were taken every 20 min.

**Figure 2 ijms-21-05083-f002:**
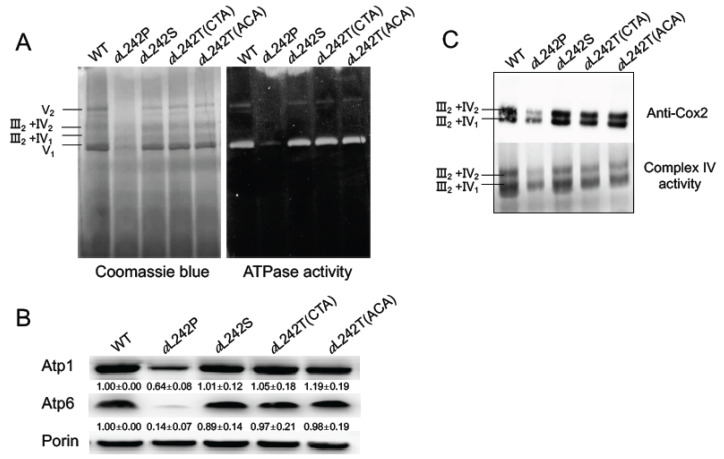
Assembly of OXPHOS complexes. For these experiments mitochondria were isolated from wild type (*WT*) and the mutant strains *a*L242P, *a*L242T and *a*L242S grown for 24 h at 28 °C in rich galactose liquid media until a density of 2-3 OD_600nm_. (**A**) Mitochondria were solubilized with 2 g of digitonin/g protein, and samples of 250 µg were separated by BN-PAGE in a 3–12% acrylamide gel. In the left panel the proteins are stained with Coomassie blue; in the right panel, F_1_F_O_ complexes are revealed in-gel by their ATPase activity. (**B**) Total mitochondrial proteins were resolved by SDS-PAGE analysis (50 µg per lane), transferred to a nitrocellulose membrane and probed with antibodies against the indicated proteins. Values are normalized to porin and expressed relative to *WT* ± standard deviation. (**C**) Mitochondria were solubilized with 10 g of digitonin/g protein, and samples of 250 µg were separated as in (A) and then transferred to a PVDF membrane. Oligomers of Complex IV associated to complex III (III_2_-IV_2_ and III_2_-IV_1_) were probed with antibodies against the Cox2 subunit of Complex IV (upper panel) and in-gel by the activity of Complex IV (lower panel). The shown gels are representative of at least 3 experiments.

**Figure 3 ijms-21-05083-f003:**
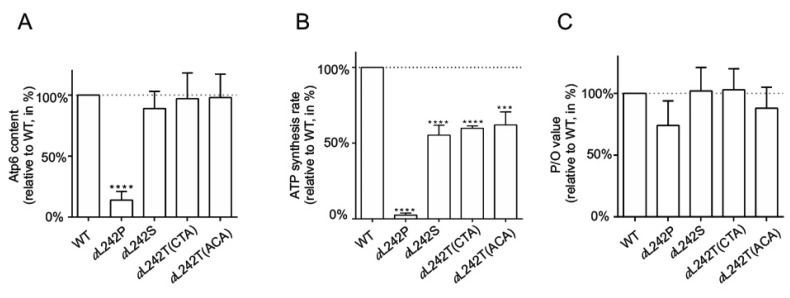
Influence of the *a*L242P, *a*L242S and *a*L242T mutations on ATP synthase accumulation and activity. The shown histograms recapitulate the relative influence on subunit *a*/Atp6 accumulation (**A**) F_1_F_O_-mediated ATP synthesis (**B**) and the yield in ATP per electron transferred to oxygen (P/O value) (**C**) of the *a*L242P, *a*L242S and *a*L242T mutations in comparison to *WT*. They were built from data reported in Figure 2B and Table 2. **** indicates a *p* < 0.0001; *** indicates a *p* < 0.001.

**Figure 4 ijms-21-05083-f004:**
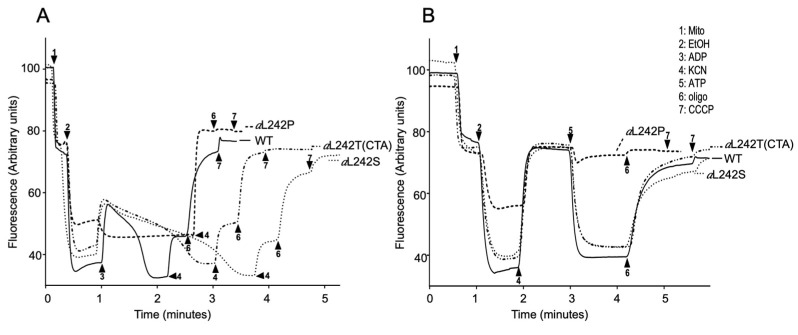
Mitochondrial membrane potential. Variations in mitochondrial ΔΨ were monitored by fluorescence quenching of Rhodamine 123, using intact, osmotically-protected, mitochondria from wild type (*WT*) and mutant strains *a*L242P, *a*L242T and *a*L242S grown for 24 h at 28 °C in rich galactose liquid media until a density of 2-3 OD_600nm_ (these mitochondrial preparations are the same as those used in Figure 2). The tracings in (**A**) show how the mitochondria respond to externally added ADP, those in (**B**) reflect ATP-driven proton-pumping by ATP synthase. The additions were 25 μg/mL Rhodamine 123, 0.15 mg/mL Mito, 10 μL EtOH, 75 μM ADP, 2 mM KCN, 0.2 mM ATP, 4 μg/mL oligo, and 4 μM CCCP. The shown fluorescence tracings are representative of at least three experiments.

**Figure 5 ijms-21-05083-f005:**
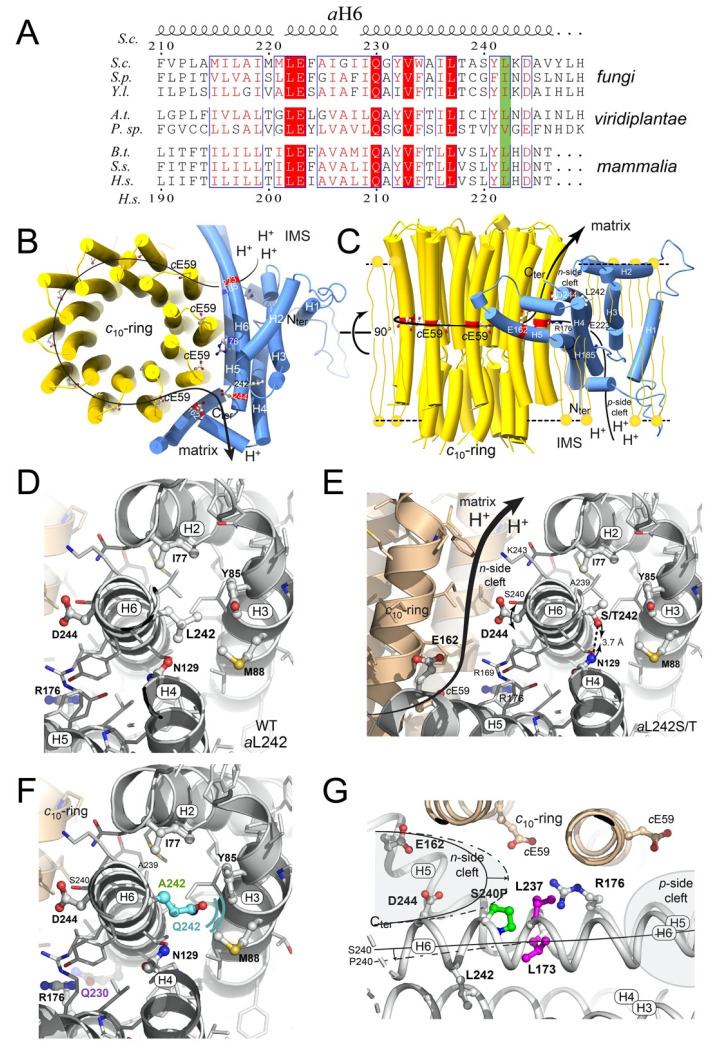
Evolutionary conservation and topology of *a*L242 and *a*L242S/T residues. (**A**) Amino-acid alignment of the C-terminal ΔΨ-helix (*a*H6) of subunits *a* from various mitochondrial origins: *Saccharomyces cerevisiae* (*S.c.*), *Schizosaccharomyces pombe* (*S.p.*), *Yarrowia lipolytica* (*Y.l.*), *Arabidopsis thaliana* (*A.t.*), *Polytomella sp* (*P.sp.*), *Bos taurus* (*B.t.*), *Sus scrofa* (*S.s.*), and *Homo sapiens* (*H.s.*). At the top and bottom, are numbered the residues in *S.c.* mature protein (i.e., without the first ten residues that are moved during assembly) and in the *H.s.* protein, respectively. Strictly conserved residues are in white on a red background while similar residues are in red on a white background with blue frames. The secondary structures of the *S.c.* protein marked above the alignment are according to [9]. Top view from the matrix (**B**) and side view (**C**) of the *c*_10_-ring and subunit *a* and the pathway along which protons are transported from the intermembrane space (IMS) to the mitochondrial matrix. The side chains of the two residues essential to this transfer (*a*R176 and *c*E59), and of residues presumed to be important for this transfer, in the n-side cleft (*a*H185, *a*E223) and in the p-side cleft (*a*E162, *a*D244), and of *a*L242 are drawn as ball and stick. (**D**) The wild type *a*L242 residue is within a 4-helix bundle (*a*H2, *a*H3, *a*H4 and *a*H6) in proximity to *a*I77, *a*Y85, *a*M88 and *a*N129. This bundle is disrupted with *a*L242P and fully preserved with *a*L242S and *a*L242T, which presumably involves the formation of a hydrogen bond between the hydroxyl group of *a*S242 or *a*T242 and the amide group of *a*N129 as depicted (**E**). The partial impairment of ATP synthase function with *a*S242 or *a*T242 is possibly caused by a local disturbance at the level of *a*D244 as indicated by the double arrowed curved trait (see Text). (**F**) In *a*L242A (green) and *a*L242Q (cyan) a cavity or clashes between neighboring helices preclude the helix bundle stability. (**G**) Schematic top view showing the probable bend of *a*H6 due to the *a*S240P mutation that should deepen the *n*-side pocket. The *a*H6 helix axis is drawn as a continuous (S240) and dashed (P240) line. For sake of clarity, only relevant side chains are depicted.

**Table 1 ijms-21-05083-t001:** Intragenic suppressors of *a*L242P.

Codon Change	Amino Acid Change	Number
Original mutant		
TTA252CCA	*a*L242P	-
Intragenic suppressors		
CCA252TCA	*a*L242S	31
CCA252ACA	*a*L242T	11
CCA252CTA	*a*L242T	3
Other possible single nucleotide substitutions not obtained		
CCA252CAA	*a*L242Q	0
CCA252CGA	*a*L242R	0
CCA252GCA	*a*L242A	0

**Table 2 ijms-21-05083-t002:** Mitochondrial respiration and ATP synthesis/hydrolysis.

Strain	Respiration Rates nmol O/min/mg				ATP Synthesis Rate nmol Pi/min/mg		ATPase Activity µmol Pi/min/mg			P/O
	NADH	NADH + ADP	NADH + CCCP	Asc/TMPD + CCCP	-oligo	+oligo	-oligo	+oligo	Inhib %	
WT	329 ± 3	619 ± 99	1548 ± 113	2865 ± 325	1531 ± 168	151 ± 30	4.7 ± 0.2	0.5 ± 0.1	88	1.21 ± 0.18
*a*L242P	96 ± 8	79 ± 11	242 ± 48	440 ± 26	142 ± 43	96 ± 34	3.3 ± 0.4	3.2 ± 0.2	4	0.89 ± 0.20
*a*L242S	192 ± 21	353 ± 23	793 ± 60	1443 ± 264	866 ± 170	170 ± 31	4.2 ± 0.1	0.6 ± 0.2	85	1.24 ± 0.19
*a*L242T(CTA)	220 ± 13	410 ± 22	949 ± 47	1704 ± 75	1056 ± 90	164 ± 32	4.2 ± 0.5	0.5 ± 0.2	88	1.25 ± 0.17
*a*L242T(ACA)	224 ± 28	462 ± 84	1133 ± 127	2101 ± 316	1066 ± 89	160 ± 18	4.0 ± 0.2	0.5 ± 0.1	87	1.06 ± 0.17

Mitochondria were isolated from wild type MR6 strain and mutant strains *a*L242P, *a*L242S and *a*L242T (with either the CTA or ACA threonine codon) grown for 5-6 generations in rich galactose medium (YPGalA) at 28 °C. Reaction mixes for assays contained 0.15 mg/mL protein, 4 mM NADH, 150 µM ADP (for respiration assays), 1 mM ADP (for ATP synthesis assays), 12.5 mM ascorbate (Asc), 1.4 mM TMPD, 4 µM CCCP, 4 µg/mL oligomycin (oligo). The values reported are averages of triplicate assays ± standard deviation. Respiration and ATP synthesis activities were detected on freshly isolated and osmotically protected mitochondria in the respiratory buffer (see Section 4, Materials and Methods). The ATPase assays were performed from mitochondrial samples that had been frozen at −80 °C in the absence of osmotic protection, in the ATPase buffer (see Materials and Methods).

**Table 3 ijms-21-05083-t003:** Genotypes and sources of the yeast strains.

Strain	Nuclear Genotype	mtDNA	Source
MR6	*MAT* *a ade2-1 his3-11,15 trp1-1 leu2-3,112 ura3-1 CAN1 arg8::HIS3*	*ρ ^+^*	[29]
RKY66	*MAT* *a ade2-1 his3-11,15 trp1-1 leu2-3,112 ura3-1 CAN1 arg8::HIS3*	*ρ^+^ ATP6-**a*L242P	[18]
RRKY66/1	*Revertant isolated from RKY66*	*ρ^+^ ATP6-**a*L242S	This study
RRKY66/2	*Revertant isolated from RKY66*	*ρ^+^ ATP6-**a*L242T (CTA)	This study
RRKY66/3	*Revertant isolated from RKY66*	*ρ^+^ ATP6-**a*L242T (ACA)	This study

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
