# Peer review of "Molecular Basis of the Pathogenic Mechanism Induced by the m.9191T>C Mutation in Mitochondrial *ATP6* Gene"

_ijms, 2020, doi:10.3390/ijms21145083_

Round 1

Reviewer 1 Report

This is an interesting study which focuses on molecular explanation of the effect of aL242P substitution on ATP synthase function. The authors had previously shown that m9191T>C mutation in mitochondrial ATP6 gene in patients with Leigh syndrome severely impairs yeast ATP synthase (aL242P ) and reduces mitochondrial ATP synthesis (Kabala et al., 2014). Saccharomyces cerevisiae was used as a model due to easy mDNA manipulation and homoplasmic clones formation. Authors tried to explain the role of the aL242 residue (corresponding to aL222 in human protein) located in C-terminus of H6 α-helice of subunit a in proton conduction.  For this, functional revertants of aL242P were isolated. Sequencing analysis revealed three types of  single-nucleotide missense substitution resulting in aL242S and aL242T mutated forms. Authors show that aP242 to Ser or Thr substitution  restored the ATP synthase assembly and stability to the level compared to WT aL242. However, the ATP synthase activity was only partially recovered. Although the yield in ATP per electron transferred to oxygen was unaffected, ATP production was slowed down (50%). Based on this observation Authors proposed that aL242 residue would probably play a role in the interaction of protons released from c-ring with aD244 residue located in C-terminus H6 α-helice near the aL242 residue.

Manuscript is well constructed and well written. The results are clearly presented and the conclusions are probably correct although I have a few points that should be discussed:

  1. Presented data shows that aL242P (TTA252CCA) substitution severely diminishes  the ability of yeast to grow on  non-fermentable carbon sources that require a functional ATP synthase. Revertants of the aP242 were isolated among which the CCA codon was converted into the TCA (serine) or ACA (threonine) or CTA (threonine). Authors notify that the true back mutation aP242L (CCA to TTA) was not represented since the expected frequency of spontaneous reversion of two nucleotides is very low  (<10-12). However, there are other possible single nucleotide substitutions like: GCA (alanine), CAA (glutamine) and CGA (arginine), which were not identified by authors as well. It raises the question: was it lost in selection? (it is possible because only max two clones per plate were verified) or this substitution impair ATP synthase even more than substitution to proline? It should be discussed.
  2. Following question 1 : Both mutants aL242S and aL242T are phenotypically similar. It is not surprising since these amino acids are similar in size, polar, quite neutral regarding mutation. According to the structural model, the aL242 residue does not face the c-ring, however, it is located within the H6 helice which directly interacts with c-ring. Based on phenotype of aS242 and aT242 mutants Authors deduced that Fo-mediated proton conduction is lowered in aS242/aT242 compared to the rate of the WT (aL242), so aL242 residue (H6) is possibly implicated in proper exposition of aD242 residue (H6), which is known to be involved in proton conduction. Authors  suggest that serine and threonine in position 242 could form hydrogen bonds with aN129 which restore stability of a subunit. However, the possible other single nucleotide substitution of CCA codon could lead to insertion of alanine, glutamine or arginine in the position of 242. The lack of these substitution mutants in the pool of revertants could shed new light on the role of L242 and should be discussed. Positively charged arginine is often responsible for formation of salt bridge with acidic amino acid, what can result in protein “over-stabilisation”– could this be critical for aD244 presentation/flexibility and/or for protein assembly ? Alanine is a small and very non-reactive amino acid- how could this substitution influence the H6 α-helice region?

Does the lack of proline to arginine or alanine or glutamine substitution whitin the pool of revertants supports Author’s conclusions? This should be discussed according to the L242 role in H6 α-helice stability and D244 residue presentation.

  1. Experimental data obtained for aL242P mutant analysis should be added to figure3.

  2. Line 27: should be m9191T>C

  3. Line 50: should be “the same”

Author Response

"Please see the aatachment"

Reviewer 2 Report

In the present manuscript, Su et al. follow up on their previous yeast model of a pathogenic mutation in the ATP6 gene, which results in a L222P substitution in the human protein. They had previously observed that the L222P mutant results in a poorly assembled and poorly functional ATP synthase. They now report that revertants of the L222P mutation to L222S or L222T rescues ATP synthase stability but only partially rescued its function. They propose that the L222 may be necessary for not only forming an alpha-helix but may have another structural function beside. While the quality of the work presented is high, the advance seems slight. If the main point is elucidate the pathogenicity of the L222P mutation, the  L222S or L222T revertants add little. Breaking the alpha-helix with a proline substitution seems a sufficient mechanism for pathogenicity. If the point is a structure-function exploration of ATP6, choosing the 222 position seems to be driven by convenience more than any rationale. I have some additional experimental concerns.

Major Concerns:

(1) In table 2 the authors observe consistent decreases in respiration and ATP synthesis rates relative to the WT strain. This is the basis of their central claim that the L242 residue has an important beyond its contribution to secondary structure. All of the revertants though are ultimately derived from a clone of the original strain, namely, alphaL242P. Could the decreased respiration and ATP synthesis rate be a clonal effect? This should be tested by generating a L242S or L242T mutant from the original WT line by mutagenesis.  

(2) Can the hypothesis that S/T forms a hydrogen bond with N129 referenced in Figure 5 be further tested? This might as they authors suggest stabilize the alpha-helix or change the orientation of D244 or both. What if residue 242 is substituted with C or N is there still rescue of stability? Is there still a slow-down of ATP synthesis.

(3) The authors imply their results suggest another function for leucine in the 222 position beyond maintaining secondary structure, but it's not clear from their discussion of Figure 5 what this might be. What is the explanation for the importance of this residue here apart from secondary structure?

Minor comments:

(1) In the introduction, NARP and MILS should be defined at first mention. The specific syndrome caused by the L222P should be described and the original clinical report cited.

(2) In table 1 it would be helpful if the authors noted which substitutions would be expected using their method of reversion (i.e., which substitutions are possible with a one nucleotide substitution). That is, reversion to leucine is not possible. What other possible substitutions are not assessed using this method?

Round 2

Reviewer 2 Report

My concerns have been addressed.